# Radiotherapy to the prostate for men with metastatic prostate cancer in the UK and Switzerland: Long-term results from the STAMPEDE randomised controlled trial

Chris C. Parker[1]*, Nicholas D. James[1], Christopher D. Brawley[2], Noel W. Clarke[3,4], Adnan Ali[3], Claire L. Amos[2], Gerhardt Attard[5], Simon Chowdhury[6], Adrian Cook[2], William Cross[7], David P. Dearnaley[1], Hassan Douis[8], Duncan C. Gilbert[2], Clare Gilson[2], Silke Gillessen[9,10], Alex Hoyle[3,4], Rob J. Jones[11], Ruth E. Langley[2], Zafar I. Malik[12], Malcolm D. Mason[13], David Matheson[14], Robin Millman[2], Mary Rauchenberger[2], Hannah Rush[2,6], J Martin Russell[11], Hannah Sweeney[2], Amit Bahl[15], Alison Birtle[16,17], Lisa Capaldi[18], Omar Din[19], Daniel Ford[8], Joanna Gale[20], Ann Henry[21], Peter Hoskin[3,22], Mohammed Kagzi[23], Anna Lydon[24], Joe M. O'Sullivan[25], Sangeeta A. Paisey[26], Omi Parikh[27], Delia Pudney[28], Vijay Ramani[3,29], Peter Robson[12], Narayanan Nair Srihari[30], Jacob Tanguay[31], Mahesh K. B. Parmar[2ᵒ], Matthew R. Sydes[2ᵒ]*, for the STAMPEDE Trial Collaborative Group[¶]

1 The Institute of Cancer Research and Royal Marsden NHS Foundation Trust, London, United Kingdom, 2 MRC Clinical Trials Unit at UCL, UCL, London, United Kingdom, 3 The Christie Hospital, Manchester, United Kingdom, 4 Salford Royal Hospitals, Manchester, United Kingdom, 5 UCL Cancer Institute, UCL, London, United Kingdom, 6 Guys and St Thomas's NHS Foundation Trust, London, United Kingdom, 7 St James's University Hospital, Leeds, United Kingdom, 8 University Hospitals Birmingham NHS Foundation Trust, Birmingham, United Kingdom, 9 Istituto Oncologico della Svizzera Italiana, EOC, Bellinzona, Switzerland, 10 Università della Svizzera Italiana, Lugano, Switzerland, 11 Institute of Cancer Sciences, University of Glasgow, Glasgow, United Kingdom, 12 Clatterbridge Cancer Centre, Liverpool, United Kingdom, 13 Cardiff University, Cardiff, United Kingdom, 14 University of Wolverhampton, Wolverhampton, United Kingdom, 15 University Hospitals Bristol NHS Trust, Bristol, United Kingdom, 16 Rosemere Cancer Centre, Lancs Teaching Hospitals, University of Manchester, Manchester, United Kingdom, 17 UCLan, Lanchashire, United Kingdom, 18 Worcestershire Acute Hospitals NHS Trust, Worcester, United Kingdom, 19 Weston Park Cancer Centre, Sheffield, United Kingdom, 20 Queen Alexandra Hospital, Portsmouth, United Kingdom, 21 University of Leeds, Leeds, United Kingdom, 22 Mount Vernon Cancer Centre, Northwood, United Kingdom, 23 The James Cook University Hospital, Middlesbrough, United Kingdom, 24 Torbay and South Devon NHS Trust, Torbay, United Kingdom, 25 Patrick G Johnston Centre for Cancer Research, Queen's University Belfast, United Kingdom, 26 Hampshire Hospitals NHS Foundation Trust, Hampshire, United Kingdom, 27 Royal Preston Hospital, Preston, United Kingdom, 28 South West Wales Cancer Centre, Swansea, United Kingdom, 29 Manchester University Hospitals NHS Trust, Manchester, United Kingdom, 30 Shrewsbury & Telford Hospitals NHS Trust, Shrewsbury, United Kingdom, 31 Velindre Cancer Centre, Cardiff, United Kingdom

ᵒ These authors contributed equally to this work.
¶ Membership of the STAMPEDE Trial Collaborative Group is listed in S4 Text.
* chris.parker@icr.ac.uk (CCP); m.sydes@ucl.ac.uk (MRS)

**Data Availability Statement:** Data will be available to successful applications for clearly specified research projects following the MRC CTU at UCL

## Abstract

### Background

STAMPEDE has previously reported that radiotherapy (RT) to the prostate improved overall survival (OS) for patients with newly diagnosed prostate cancer with low metastatic burden, but not those with high-burden disease. In this final analysis, we report long-term findings on

standard data sharing processes: https://www.mrcctu.ucl.ac.uk/our-research/other-research-policy/data-sharing/ Discussion with the trial team is encouraged to determine whether the relevant data to support the application are available. Email to: mrcctu.datareleaserequest@ucl.ac.uk.

**Funding:** Research support for this comparison and other comparisons in the STAMPEDE protocol was awarded by Cancer Research UK (CRUK_A12459) www.cancerresearchuk.org (for this comparison, co-authors CCP, DPD, MDM, MKBP, MR, MRS, NDJ; and additionally for other comparisons DG, DM, GA, REL, RM, WC); Medical Research Council (MRC_MC_UU_12023/25, MC_UU_00004/01 and MC_UU_00004/02) www.ukri.org/councils/mrc (to authors MKBP, MRS, REL); and Swiss Group for Clinical Cancer Research, www.sakk.ch (to co-author SG). Other research support for the STAMPEDE protocol was awarded by Astellas www.astellas.com, Clovis Oncology www.clovisoncology.com, Janssen www.janssen.com, Novartis www.novartis.com, Pfizer www.pfizer.com, Sanofi-Aventis www.sanofi.com. CCP, DPD and NDJ are supported by the National Institute for Health Research (NIHR) Biomedical Research Centre at The Royal Marsden NHS Foundation Trust and the Institute of Cancer Research, London. The funders had no role in study design, data collection and analysis, decision to publish, or preparation of the manuscript.

**Competing interests:** I have read the journal's policy and the authors of this manuscript have the following competing interests: CCP reports personal fees from Bayer, personal fees from Janssen, personal fees from Clarity Pharmaceuticals, personal fees from Myovant, personal fees from ITM Oncologics, outside the submitted work NDJ received research funding to the institution from Astellas, Astra Zeneca &Janssen; receipt of honoraria/fees on the advisory board for Astra Zenenca, Clovis, Janssen, Merck, Novartis & Sanofi; received fees as a speaker for Bayer & Novartis NWC received honoraria from Astellas & Janssen; took a consulting/advisory role for Astellas, Janssen, Ferring, Bayer & Sanofi; was paid speakers fees b Janssen & Astellas; received funding for the institution from Astra Zeneca; received meeting and travel expenses from Janssen, Astellas, Sanofi, Astra Zeneca, Ferring & Ipsen GA reports personal fees from Sanofi Aventis, during the conduct of the study; personal fees and non-financial support from Astellas, personal fees and non-financial support from Medivation, personal fees from Novartis, personal fees from Millennium Pharmaceuticals, personal fees and non-financial

the primary outcome measure of OS and on the secondary outcome measures of symptomatic local events, RT toxicity events, and quality of life (QoL).

## Methods and findings

Patients were randomised at secondary care sites in the United Kingdom and Switzerland between January 2013 and September 2016, with 1:1 stratified allocation: 1,029 to standard of care (SOC) and 1,032 to SOC+RT. No masking of the treatment allocation was employed. A total of 1,939 had metastatic burden classifiable, with 42% low burden and 58% high burden, balanced by treatment allocation. Intention-to-treat (ITT) analyses used Cox regression and flexible parametric models (FPMs), adjusted for stratification factors age, nodal involvement, the World Health Organization (WHO) performance status, regular aspirin or nonsteroidal anti-inflammatory drug (NSAID) use, and planned docetaxel use. QoL in the first 2 years on trial was assessed using prospectively collected patient responses to QLQ-30 questionnaire.

Patients were followed for a median of 61.3 months. Prostate RT improved OS in patients with low, but not high, metastatic burden (respectively: 202 deaths in SOC versus 156 in SOC+RT, hazard ratio (HR) = 0·64, 95% CI 0.52, 0.79, $p < 0.001$; 375 SOC versus 386 SOC+RT, HR = 1.11, 95% CI 0.96, 1.28, $p = 0·164$; interaction $p < 0.001$). No evidence of difference in time to symptomatic local events was found. There was no evidence of difference in Global QoL or QLQ-30 Summary Score. Long-term urinary toxicity of grade 3 or worse was reported for 10 SOC and 10 SOC+RT; long-term bowel toxicity of grade 3 or worse was reported for 15 and 11, respectively.

## Conclusions

Prostate RT improves OS, without detriment in QoL, in men with low-burden, newly diagnosed, metastatic prostate cancer, indicating that it should be recommended as a SOC.

## Trial registration

ClinicalTrials.gov NCT00268476, ISRCTN.com ISRCTN78818544.

## Author summary

### Why was this study done?

- Prostate cancer is the most common cancer in males.

- Radiotherapy (RT) to the prostate is widely used as a radical treatment for nonmetastatic prostate cancer.

- A comparison was added to the STAMPEDE protocol to assess whether RT to the prostate would also be helpful for males with metastatic prostate cancer. A benefit in survival was targeted.

support from Abbott Laboratories, personal fees and non-financial support from Essa Pharmaceuticals, personal fees and non-financial support from Bayer Healthcare Pharmaceuticals, personal fees from Takeda, grants from AstraZeneca, grants from Arno Therapeutics, grants from Innocrin Pharma, grants, personal fees and non-financial support from Janssen, personal fees from Veridex, personal fees and non-financial support from Roche/Ventana, personal fees and non-financial support from Pfizer, personal fees from The Institute of Cancer Research (ICR), outside the submitted work; and The Institute of Cancer Research (ICR) receives royalty income from abiraterone I receive a share of this income through the ICR's Rewards to Discoverers Scheme SC received consulting fees from Telix, remedy & Huma; received payment for speaker fees and/or manuscript writing and/or educational events from Astra Zeneca, Novartis/AAA, Clovis, Janssen, Bayer, Pfizer, Beigene & Astellas; they were a member of the data safety monitoring/advisory board for Astra Zeneca, Novartis/AAA, Clovis, Janssen, Bayer, Pfizer, Beigene & Astellas DPD received payment to the institution from C33589/A19727 Advances in Physics for Precision Radiotherapy; previous employer, The Institute of Cancer Research receives loyalty income from abiraterone, receives personal share of this income through ICR's Rewards to Discoverer's Scheme; honoraria for consultancy from Janssen; EP1933709B1 – Location and Stabilisation Device., European patent issued, Pending in Canada and India SG reports personal fees from Orion, personal fees from Janssen Cilag, personal fees from ProteoMedix, personal fees from Amgen, personal fees from MSD, other from Tolero Pharmaceuticals, other from Astellas Pharma, other from Janssen, other from MSD Merck Sharp&Dome, other from Bayer, other from Roche, other from Pfizer, other from Telixpharma, other from Amgen, other from Bristol-Myers Squibb, other from AAA International SA, other from Orion, other from Silvio Grasso Consulting, from Tolremo, outside the submitted work; In addition, Gillessen has a patent WO2009138392 issued and Menarini Silicon Biosystems (Advisory Board 2019) - not compensated Aranda (Advisory Board 2019) - not compensated RJJ received research funding to the institution from Bayer, Astellas & Pfizer; received honoraria on the advisory board for Janssen, Astellas, Bayer, Pfizer; received speaker fees from Janssen, Astellas, Bayer & Pfizer REL received an institutional grant from the MRC CG received research funding to the institution from Janssen, Clovis Oncology, Sanofi, Astellas, Medical Research Council & Cancer

- The trial previously reported a clinically relevant, statistically significant overall survival (OS) benefit for patients with a low metastatic burden but not for men with a high metastatic burden.

- This long-term analysis assesses survival with substantially longer follow-up and more events and looked also at complications of local disease.

### What did the researchers do and find?

- A randomised controlled trial of adding RT to the prostate to standard of care (SOC) was incorporated into the STAMPEDE protocol.

- More than 2,000 patients joined the comparison between 2013 and 2016.

- The data set was frozen in 2021 and analysed using standard methods.

- There was a clear improvement in survival with prostate RT in the low metastatic burden group.

- There was no improvement in survival with prostate RT in the high metastatic burden group.

- Symptomatic local progression and the need for later local intervention were improved with RT in the low metastatic burden group.

- In the low metastatic burden group, the improvement with RT was similar whether the RT was given with a daily schedule (over 4.5 weeks) or a weekly schedule (over 6 weeks).

- The adverse effects of RT were manageable without any impact on long-term quality of life (QoL).

### What do these findings mean?

- Prostate RT is a relatively cheap, widely accessible, and well-tolerated treatment.

- Prostate RT is indicated in patients with newly diagnosed prostate cancer with a low metastatic burden.

- RT to the prostate is not routinely indicated for patients with a high metastatic burden.

### Introduction

Prostate radiotherapy (RT) is recommended for men with newly diagnosed, low-burden, metastatic prostate cancer, but not for men with high-burden disease [1]. This recommendation is based largely on the initial results of the STAMPEDE trial, reported in 2018 [2]. In this randomised controlled trial of 2,061 men with newly diagnosed metastatic prostate cancer, prostate RT improved overall survival (OS) for men with low metastatic burden (hazard ratio [HR]

Research UK DF received speaker fees and/or manuscript writing and/or educational events from BMS, IPSEN, EUSA, Pfizer, ESAI; they received travel expenses from Janssen & IPSEN MDM is an advisory board member for Endocyte & Clovis AB received payment for lecture/presentation/speaker bureau/manuscript writing or educational event from Boston Scientific AJB received speaker fees and travel support from Janssen DF received payment for lectures for Janssen, Pfizer & BMS; support for attending conferences/meetings from Genisiscare & BMS AMH received research grants from CRUK and NIHR; support attending meetings from the European Association of Urologists; is a member of the European Association of Urologists & the Prostate Cancer Guidelines Group MK received travel, accommodation and conference fees as expenses from Bayer, travel and accommodation fees for Prostate cancer summits from Janssen AL received expenses for attending meetings and/or travel from Astellas, Bayer, BMS & MSD JMOS received speaker fees from AAA, Astellas, Bayer, Janssen, Novartis, Sanofi and participated as an advisory board member and/or member of the data safety monitoring board for AAA, Astellas, Bayer, Janssen, Novartis & Sanofi NNS received travel/meeting payments from Janssen JT received support for conference attendance from Janssen, Roche & Bayer; participated on the advisory board for Astra Zeneca, Astellas & Bayer MKBP received research funding to the Unit he directs from Acoria Pvt Ltd, Akagera, Amgen, Aspirin Foundation, Astellas, AstraZeneca, Baxter, Bayer, BMS US, Bri-Bio, Cepheid, Cipla, Clovis Inc, CSL Behring, Eli-Lilly, Emergent Biosolutions, Gilead Sciences, GlaxoSmithKline, Grifols, Janssen Products LP, Janssen-Cilag, Johnson & Johnson, Micronoma, Modus Theraputics, Mylan, Novartis, Pfizer, Sanofi, Serum Institute of India, Shionogi, Synteny Biotechnology, Takeda, Tibotec, Transgene, ViiV Healthcare, Virco and Xenothera MRS received research funding to the institution from Astellas, Clovis, Janssen, Novartis, Pfizer, Sanofi-Aventis; received speaker fees from Lilly Oncology & Janssen; independent member of data monitoring committees. All other authors have nothing to declare.

**Abbreviations:** ADT, androgen deprivation therapy; AE, adverse event; CONSORT, Consolidated Standards of Reporting Trials; CTCAE, Common Terminology Criteria for Adverse Events; FPM, flexible parametric model; GnRH, gonadotrophin-releasing hormone; HR, hazard ratio; IQR, interquartile range; ITT, intention-to-treat; LIFS, local intervention–free survival; NSAID, nonsteroidal anti-inflammatory drug; OS, overall

0.68, 95% CI 0.52 to 0.90; $p = 0.007$), with no evidence of a meaningful effect on survival in men with high metastatic burden (HR 1.07, 95% CI 0.90 to 1.28; $p = 0.420$). That initial analysis, triggered by a preplanned number of events, was done after a median follow-up of 37 months and was based on 761 events. Here, we report the final analysis of OS, with an additional 2 years follow-up.

We hypothesised that prostate RT would reduce the complications of local disease progression, such as urinary or bowel obstruction. If so, this could benefit men with metastatic disease, regardless of disease burden. Here, we report data on freedom from local interventions (e.g., urinary catheter, ureteric stents, nephrostomies, and colostomy).

Any benefits of prostate RT need to be weighed against the risk of treatment-related adverse events (AEs). We report, for the first time, data from the trial on quality of life (QoL).

The trial was stratified according to the choice of 1 of 2 RT dose-fractionation schedules, nominated prior to randomisation; 36 Gy in 6 fractions over 6 weeks, or 55 Gy in 20 fractions over 4 weeks. The 2 schedules were chosen in the expectation that they would be similarly effective. With the benefit of additional follow-up, and more events in the final analysis, we have tested for any differential impact on OS by choice of RT schedule.

## Methods

### Study participants

Eligible patients had prostate cancer that was newly diagnosed, with no previous radical treatment, had metastatic disease confirmed on a bone scintigraphic scan and soft tissue imaging, and were within 12 weeks after starting androgen deprivation therapy (ADT). Patients were required to have no contraindications to RT and no clinically significant cardiovascular history. Participants were recruited at secondary care sites in the UK and Switzerland.

The trial was registered as NCT00268476 (ClinicalTrials.gov) and ISRCTN78818544 (ISRCTN.com). The trial was done in accordance with Good Clinical Practice guidelines and the Declaration of Helsinki and had relevant ethics (West Midlands–Edgbaston Research Ethics Committee) and regulatory approvals. All patients gave written informed consent. The rationale and design, including sample size calculations, have been described previously [2,3]. Full details are in the protocol at www.stampedetrial.org.

### Procedures

All patients received lifelong hormone therapy as gonadotrophin-releasing hormones (GnRHs) agonists or antagonists or orchidectomy. In addition, docetaxel was permitted after it became available for this setting in the UK. Docetaxel, when used, was given as six 3 weekly cycles of 75mg/m$^2$ with or without prednisolone 10 mg daily.

External beam RT to the prostate was given as 1 of 2 schedules nominated prior to randomisation: 36 Gy in 6 consecutive weekly fractions of 6 Gy or 55 Gy in 20 daily fractions of 2.75 Gy over 4 weeks. Treatment was given with the patient supine, with a full bladder and an empty rectum. The planning target volume consisted of the prostate only with an 8-mm margin posteriorly and a 10-mm margin elsewhere. RT was to commence as soon as practicable after randomisation, and, if the patient was having docetaxel as part of standard of care (SOC), within 3 to 4 weeks after the last docetaxel dose.

Patients were followed up 6 weekly until 6 months after randomisation, 12 weekly to 2 years, 6 monthly to 5 years, and then annually. Toxicities and symptoms were reported at regular follow-up visits or when an AE was categorised as "serious." These were graded with Common Terminology Criteria for Adverse Events (CTCAE) v4·0. Separately, bowel and bladder adverse effects during RT and long-term possible RT effects were recorded using the RTOG

survival; PH, proportional hazard; QoL, quality of life; RMST, restricted mean event-free ("survival") time; RT, radiotherapy; SLEFS, symptomatic local event-free survival; SOC, standard of care; WHO, World Health Organization.

scale [4]. Participants were asked to complete the EORTC QLQ-C30 at each scheduled follow-up appointment.

Metastatic burden at randomisation was evaluated retrospectively through central imaging review of whole body scintigraphy and computerized tomography (CT) or MRI staging scans. Metastatic burden was classified according to the definition used in the CHAARTED trial [5] as either high (polymetastatic; ≥4 bone metastases with ≥1 outside the vertebral bodies or pelvis and/or visceral metastases) or low (oligometastatic). Patients with only lymph node metastases, in the absence of bone or visceral disease, were therefore classified as low metastatic burden regardless of the number of nodal metastases.

## Randomisation and masking

Patients were randomised centrally using a computerised algorithm, developed and maintained by the trials unit. Minimisation with a random element of 20% was used (80% probably of allocation to a minimising treatment), stratifying for hospital, age at randomisation (<70 versus ≥70 years), nodal involvement (negative versus positive versus indeterminate), the World Health Organization (WHO) performance status (0 versus 1 or 2), planned form of ADT (orchidectomy versus LHRH (leuteinising hormone-releasing hormone) agonist versus LHRH antagonist versus dual androgen blockade), and regular aspirin or nonsteroidal anti-inflammatory drug (NSAID) use (yes or no). Planned docetaxel use was added as a stratification factor after use was permitted as part of SOC. Allocation was 1:1 to SOC only or SOC+RT. There was no blinding to treatment allocation.

## Primary and secondary outcomes

The primary efficacy outcome measure was OS, defined as time from randomisation to death from any cause. Secondary outcomes for this long-term efficacy analysis included local intervention–free survival (LIFS)—consisting of time from randomisation to the first report on case report forms of TURP, ureteric stent, surgery for bowel obstruction, urinary catheter, nephrostomy, colostomy, death from prostate cancer—and symptomatic local event-free survival (SLEFS), comprising any of these LIFS events or acute kidney injury, urinary tract infection, or urinary tract obstruction. Cause of death was determined by the site investigator, with some cases reclassified as prostate cancer death according to predefined criteria which suggested this to be the likely cause. Patients without the event of interest were censored at the time last known to be event free. QoL analyses focused on Global QoL % and QLQ-30 Summary Score %, as derived from patient reports at scheduled assessment time points in the first 2 years after randomisation (see **S2 Text**).

## Statistical analysis

The primary outcome measure, OS, was assessed across all patients and separately within patient subgroups characterised by baseline metastatic burden (low versus high) and nominated RT schedule (daily versus weekly).

Standard survival analysis methods were used to analyse time-to-event data in Stata v16.1 (College Station, Texas, United States of America). A nonparametric stratified log-rank test was used to assess any difference in survival between treatment groups; this was stratified across the minimisation factors used at randomisation (except hospital and planned form of hormone therapy) plus protocol-specific time periods defined by other arms recruiting to STAMPEDE or changes to SOC which could affect the population being randomised. Cox proportional hazards (PHs) regression models adjusting for the same stratification factors and stratified by time period were used to estimate relative treatment effect; a HR less than 1·00

favoured the research arm. Unadjusted estimates of treatment effect are also presented. Flexible parametric models (FPMs) were fitted with degrees of freedom (5.5) and adjusted for stratification factors and time periods [6]. Medians and 5-year survival estimates are presented from the FPM fitted to the data. Kaplan–Meier curves, using the KMunicate format [7], show estimated survival over time. Following the fitting of Cox models, the PHs assumption was tested using a global Grambsch–Therneau test with log-transformed time; restricted mean event-free ("survival") time (RMST) was emphasised in the presence of nonproportionality, using a t-star of 91 months as determined by the Royston and Parmar method [6]. Cox and Fine and Gray regression models [8] were used for cause-specific and competing risk analyses, respectively; competing risks were non-prostate cancer–related death for prostate cancer–specific survival and death from any cause for SLEFS and LIFS. Evidence for different treatment effect across subgroups was assessed using the likelihood ratio *p*-value for an interaction term added to the relevant adjusted Cox/FPM model, or Wald test *p*-value from Fine and Gray model. Sensitivity analyses of local event outcomes examined the impact of excluding death from prostate cancer and a competing risks approach with death from any cause specified as a competing event. All tests are presented as 2 sided, with 95% CIs and the relevant *p*-value.

Median follow-up was estimated using the Kaplan–Meier method with reverse censoring on death. All patients were included in the efficacy and QoL analyses according to allocated treatment on an intention-to-treat (ITT) basis; sensitivity analyses exclude patients who did not explicitly fulfill all of the eligibility criteria. AE data are shown for the safety population, in patients with at least 1 follow-up assessment and analysed according to whether RT was received within 1 year of randomisation (SOC+RT) or not (SOC).

Analyses of the QoL outcomes included partly conditional and composite approaches, building on the approaches previously used in the trial [9]. For the former, missing values were multiply imputed using observed data using chained equations. Imputed values for assessments dating after a patient had died were restored to missing. Generalised estimating equations with an independence correlation matrix were used to estimate the expected value of the outcome for each treatment arm at each assessment time point. For the composite approach, observations following the death of a patient were set to 0% (corresponding to the lowest possible QoL state). Mixed linear regression with random intercept and slope (with unstructured correlation specification) was used to model the outcome. Additional cross-sectional analyses estimated the difference in average QoL associated with treatment allocation in patients alive and with data available at a given assessment time point, controlling for baseline state.

This trial is reported per the Consolidated Standards of Reporting Trials (CONSORT; see **S3 Text**).

## Results

### Patients

Between January 22, 2013 and September 2, 2016, 2,061 patients were randomised from 117 hospitals in UK and Switzerland: 1,029 to SOC and 1,032 to SOC+RT. The data set was frozen on March 17, 2021 and included information up to November 30, 2020. **Fig 1** shows the CONSORT flow diagram for analyses presented in this paper. **Table 1** shows baseline characteristics balanced across the allocated treatment groups. **Table A in S1 Text** shows baseline characteristics in 1,939 (94%) patients who were evaluable for disease burden, 819 (40%) with low- and 1,120 (54%) with high-burden disease.

Median duration of follow-up was 61.3 months (interquartile range [IQR] = 53.8 to 73.1) and was similar in both treatment groups: SOC 61.0 (IQR = 53.8 to 72.6) and SOC+RT 61.6 (IQR = 53.8 to 73.1).

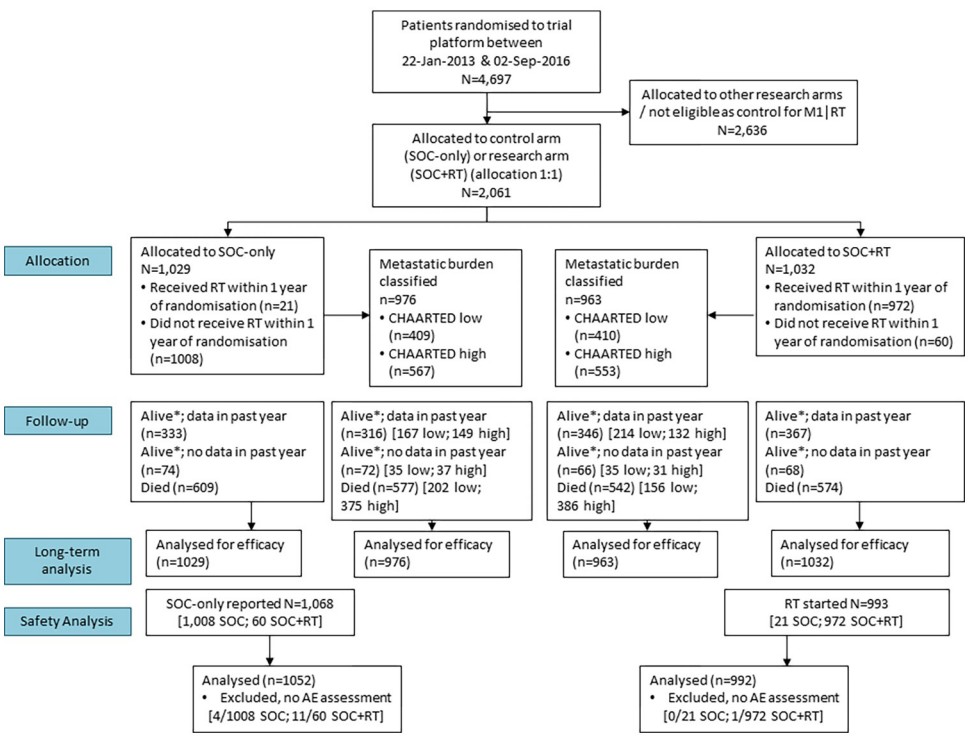

**Fig 1. CONSORT diagram.** AE, adverse event; CONSORT, Consolidated Standards of Reporting Trials; RT, radiotherapy to the prostate, SOC, standard of care. *Alive, no withdrawal of permission for continued data collection.

## OS by allocated treatment and metastatic burden

A total of 1,183 deaths were reported, 609 in patients allocated to SOC and 574 in those allocated to SOC+RT (**Fig 2**, **Table 2**).

In the low metastatic burden group, 358 had died: 202/409 SOC and 156/410 SOC+RT. Median survival was 63.6 months for SOC and 85.5 months for SOC+RT (5-year survival 53% versus 65%); adjusted HR = 0.64 (95% CI 0.52 to 0.79; $p < 0.001$ [$p = 0.00004$]) (**Fig 3**, **Table 2**). There was no evidence of non-PHs.

In the high-burden disease group, 761 had died: 375/567 SOC and 386/553 SOC+RT. Median survival was 41.2 months in SOC and 38.8 months in SOC+RT (5-year survival 35% versus 30%): adjusted HR = 1.11 (95% CI 0.96 to 1.28; $p = 0.164$) (**Fig 4**, **Table 2**). There was no evidence of non-PHs.

There was clear evidence of differential treatment effect according to metastatic burden: interaction test $p < 0.001$ [$p = 0.00005$].

Similar results were obtained from cause-specific and competing risk analyses (**Table 2**). A participant audit found 36 (<2%) patients with baseline data or documented protocol deviation inconsistent with the comparison's full eligibility criteria. Sensitivity analyses found no impact from excluding these patients (**Tables B and C in S1 Text**). Analysis of time from randomisation to reported second-line treatments indicates no confounding of RT treatment effect on OS by postprogression abiraterone or enzalutamide therapy (**S10 and S11 Figs**).

## Exploration of OS by elected RT schedule

In 980 patients nominated prior to randomisation for weekly RT, 575 had died: 282/482 SOC and 293/498 SOC+RT. Median survival was 52.2 months for SOC and 49.9 months for

**Table 1. Baseline characteristics of all patients in the comparison.**

| Characteristic | | SOC (*n* = 1,029) | SOC+RT (*n* = 1,032) |
|---|---|---|---|
| Age at randomisation (years) | Median (IQR) | 68 (63 to 73) | 68 (63 to 73) |
| | Range | 37 to 86 | 45 to 87 |
| WHO performance status | 0 | 732 (71%) | 734 (71%) |
| | 1 to 2 | 297 (29%) | 298 (29%) |
| Pain from prostate cancer | Absent | 826 (81%) | 855 (83%) |
| | Present | 198 (19%) | 172 (17%) |
| | *Missing* | 5 | 5 |
| Previous notable health issues | Myocardial infarction | 67 (7%) | 58 (6%) |
| | Cerebrovascular disease | 29 (3%) | 32 (3%) |
| | Congestive heart failure | 5 (<1%) | 8 (1%) |
| | Angina | 46 (4%) | 52 (5%) |
| | Hypertension | 408 (40%) | 444 (43%) |
| T-category at randomisation | T0 | 0 (0%) | 1 (<1%) |
| | T1 | 12 (1%) | 12 (1%) |
| | T2 | 84 (9%) | 89 (9%) |
| | T3 | 585 (62%) | 603 (63%) |
| | T4 | 260 (28%) | 247 (26%) |
| | TX | 88 | 80 |
| N-category at randomisation | N0 | 345 (36%) | 344 (36%) |
| | N+ | 620 (64%) | 620 (64%) |
| | NX | 64 | 68 |
| Metastatic burden | Low metastatic burden* | 409 (42%) | 410 (43%) |
| | High metastatic burden | 567 (58%) | 553 (57%) |
| | Not classified | 53 | 69 |
| Sites of metastases | Bone | 919 (89%) | 917 (89%) |
| | Liver | 23 (2%) | 19 (2%) |
| | Lung | 42 (4%) | 48 (5%) |
| | Distant lymph nodes | 295 (29%) | 304 (29%) |
| | Other | 35 (3%) | 32 (3%) |
| Gleason sum score | < = 7 | 173 (17%) | 175 (18%) |
| | 8 to 10 | 826 (83%) | 820 (82%) |
| | Unknown | 30 | 37 |
| PSA pre-ADT (ng/ml) | Median (IQR) | 98 (30 to 316) | 97 (33 to 313) |
| | Range | 1 to 20,590 | 1 to 11,156 |
| Time from diagnosis (days) | Median (IQR) | 73 (55 to 94) | 73 (55 to 93) |
| | *Missing* | 1 | 2 |
| Days from starting hormones | Median (IQR) | 53 (35 to 70) | 55 (34 to 70) |
| | Range | -3 to 84 | 0 to 86 |
| | *Missing* | *17* | *13* |
| Planned for SOC docetaxel | No | 845 (82%) | 849 (82%) |
| | Yes | 184 (18%) | 183 (18%) |
| Nominated RT schedule | 36 Gy/6 f over 6 weeks | 482 (47%) | 498 (48%) |
| | 55 Gy/20 f over 4 weeks | 547 (53%) | 534 (52%) |

*Note: One patient classified with low-burden disease was subsequently restaged as nonmetastatic by the randomising site. They remain in the low metastatic burden subgroup for this analysis.

ADT, androgen deprivation therapy; IQR, interquartile range; PSA, prostate specific antigen; RT, radiotherapy to the prostate; SOC, standard of care; WHO, World Health Organization.

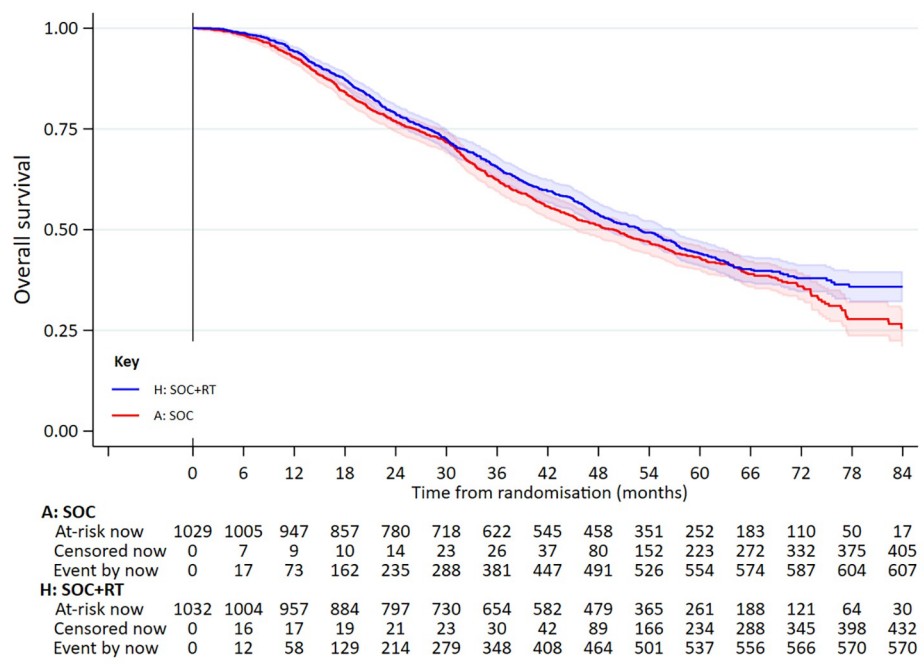

**Fig 2. OS in all patients.** Adjusted HR = 0.90 (95% CI 0.81 to 1.01; $p$ = 0.081). HR, hazard ratio; OS, overall survival; RT, radiotherapy to the prostate; SOC, standard of care.

SOC+RT (5-year survival: 44% versus 42%); adjusted HR = 1.00 (95% CI 0.85 to 1.18); $p$ = 0.974 (**Fig 5**, **Table 2**). In 1,081 patients nominated prior to randomisation for daily RT, 608 died: 327/547 SOC and 281/534 SOC+RT. Median survival was 47.8 months in SOC and 55.5 months in SOC+RT (5-year survival 41% versus 47%); adjusted HR = 0.83 (95% CI 0.71 to 0.97; $p$ = 0.022) (**Fig 6**, **Table 2**). There was no good evidence of interaction in the treatment effect by RT schedule: interaction $p$ = 0.088.

Given that RT improved OS in the low metastatic burden patients, RT schedule was further explored in this subgroup. In 360 patients nominated for weekly RT, 162 had died: 94/190 SOC and 68/170 SOC+RT; adjusted HR = 0.67 (95% CI 0.49 to 0.93; $p$ = 0.015 [$p$ = 0.0155]). In 459 patients nominated for daily RT, 196 had died: 108/219 SOC and 88/240 SOC+RT; adjusted HR = 0.62 (95% CI 0.47–0.83; $p$ = 0.001 [$p$ = 0.00112]). There was no good evidence of interaction in the treatment effect by RT schedule: interaction $p$ = 0.732.

## SLEFS by allocated treatment

A total of 1,209 (59%) patients were reported as experiencing at least 1 symptomatic local event: 608 SOC and 601 SOC+RT. In 789 cases (400 SOC and 389 SOC+RT), death from prostate cancer was the only event recorded. **Table 3** summarises the reported incidence of each type of event. There was no evidence of a difference in time to first reported event by treatment arm: adjusted HR = 1.00 (95% CI 0.90 to 1.13; $p$ = 0.931); median symptomatic local event–free survival 43.8 months SOC, 43.3 months SOC+RT (5-year SLEFS survival 39% versus 40%) (**S1 Fig**, **Table 2**).

A total of 1,086 (53%) patients had 1 or more local intervention events reported, 556 SOC and 530 SOC+RT, of which death from prostate cancer was the only event in 78% and 81% of cases. Median local intervention event–free survival was 51.1 months in SOC and 53.6 months

**Table 2. Summary of estimated treatment effect for main outcome measures: all patients and metastatic burden subgroups.**

| Outcome measure | Patient group | Adjusted HR~ | Unadjusted HR^ | Event free at 5 years+ | | RMST+ | | |
|---|---|---|---|---|---|---|---|---|
| | | | | SOC | SOC+RT | SOC | SOC+RT | Difference |
| **OS** | All patients | 0.90 (0.81 to 1.01) | 0.90 (0.81 to 1.01) | 42% | 45% | 52.9 | 55.5 | 2.5 (−0.2 to 5.2) |
| | Low metastatic burden | 0.64 (0.52 to 0.79) | 0.66 (0.54 to 0.82) | 53% | 65% | 60.6 | 69.0 | 8.4 (4.5 to 12.2) |
| | High metastatic burden | 1.11 (0.96 to 1.28) | 1.08 (0.94 to 1.25) | 35% | 30% | 47.7 | 45.5 | −2.2 (−5.7 to 1.2) |
| | Weekly RT (36 Gy/6 f) | 1.00 (0.85 to 1.18) | 1.01 (0.86 to 1.19) | 44% | 42% | 53.9 | 53.6 | −0.3 (−3.4 to 2.8) |
| | *(Low metastatic burden)* | *0.67 (0.49 to 0.93)* | *0.71 (0.52 to 0.97)* | *54%* | *64%* | *61.3* | *68.2* | *6.9 (0.6 to 13.2)* |
| | *(High metastatic burden)* | *1.22 (0.99 to 1.50)* | *1.19 (0.97 to 1.46)* | *37%* | *29%* | *48.9* | *44.5* | *−4.3 (−9.6 to 0.9)* |
| | Daily RT (55 Gy/20 f) | 0.83 (0.71 to 0.97) | 0.81 (0.69 to 0.95) | 41% | 47% | 52.2 | 57.2 | 5.0 (1.1 to 8.9) |
| | *(Low metastatic burden)* | *0.62 (0.47 to 0.83)* | *0.63 (0.48 to 0.84)* | *52%* | *66%* | *59.9* | *69.5* | *9.6 (4.0 to 15.2)* |
| | *(High metastatic burden)* | *1.02 (0.83 to 1.25)* | *0.99 (0.81 to 1.21)* | *33%* | *32%* | *46.8* | *46.6* | *−0.2 (−4.5 to 4.0)* |
| **Prostate cancer–specific survival**[*] | All patients | 0.92 (0.81 to 1.04) | 0.92 (0.81 to 1.04) | 49% | 51% | 57.6 | 59.5 | 1.9 (−1.1 to 5.0) |
| | Low metastatic burden | 0.62 (0.49 to 0.79) | 0.64 (0.50 to 0.81) | 62% | 72% | 65.7 | 73.7 | 8.0 (4.0 to 12.0) |
| | High metastatic burden | 1.12 (0.96 to 1.31) | 1.10 (0.94 to 1.28) | 41% | 35% | 51.8 | 49.0 | −2.8 (−6.6 to 1.0) |
| **SLEFS**[#] | All patients | 1.00 (0.90 to 1.13) | 1.00 (0.90 to 1.12) | 39% | 40% | 49.2 | 48.9 | −0.3 (−3.5 to 2.8) |
| | Low metastatic burden | 0.72 (0.59 to 0.88) | 0.73 (0.60 to 0.90) | 46% | 58% | 54.5 | 61.8 | 7.2 (2.5 to 11.9) |
| | High metastatic burden | 1.23 (1.06 to 1.42) | 1.21 (1.05 to 1.40) | 33% | 26% | 45.1 | 39.4 | −5.8 (−9.7 to −1.9) |
| **LIFS**[#] | All patients | 0.94 (0.83 to 1.06) | 0.93 (0.83 to 1.05) | 44% | 47% | 53.5 | 55.1 | 1.6 (−1.5 to 4.7) |
| | Low metastatic burden | 0.62 (0.49 to 0.77) | 0.63 (0.50 to 0.78) | 54% | 67% | 59.7 | 69.1 | 9.5 (5.2 to 13.8) |
| | High metastatic burden | 1.18 (1.01 to 1.37) | 1.16 (1.00 to 1.34) | 38% | 32% | 49.0 | 44.7 | −4.4 (−8.4 to −0.4) |

Note: HR and RMST difference are for SOC+RT relative to SOC.

[*]Cause-specific treatment × metastatic burden interaction test $p < 0.001$ [$p = 0.0000977$]. Competing risks analysis: overall adjusted sub-HR = 0.93 (95% CI 0.82 to 1.05; $p = 0.260$); low-burden adjusted sub-HR = 0.66 (95% CI 0.52 to 0.83; $p = 0.001$); high-burden adjusted sub-HR = 1.11 (95% CI 0.95 to 1.29; $p = 0.189$); treatment × metastatic burden interaction test $p < 0.001$ [$p = 0.000350$].

[#]SLEFS: treatment × metastatic burden interaction test $p < 0.001$ [$p = 0.0000314$]. LIFS interaction $p < 0.001$ [$p = 2.53 \times 10^{-6}$].

~Estimates from Cox models adjusting for age, nodal involvement, WHO performance status, regular aspirin or NSAID use, and planned SOC docetaxel at randomisation, stratified by randomisation time period.

^Estimates from unadjusted, unstratified Cox models.

+Survival probabilities and RMST estimates are taken from FPMs with t-star = 91 months.

HR, hazard ratio; LIFS, local intervention–free survival; NSAID, nonsteroidal anti-inflammatory drug; RMST, restricted mean event-free ("survival") time; RT, radiotherapy to the prostate; SLEFS, symptomatic local event–free survival; SOC, standard of care; WHO, World Health Organization.

in SOC+RT (5-year survival 44% versus 47%); adjusted HR = 0.94 (95% CI 0.83 to 1.06; $p = 0.286$) (S2 Fig, Table 2, Table D in S1 Text). Table E in S1 Text presents the results of sensitivity analyses.

## AEs by allocated treatment

Urinary-related late AEs of grade 3 were reported for 20 (2%) patients who received RT within 1 year after randomisation; 10 (2%) were planned for weekly and 10 (2%) for daily treatment; no grade 4 or 5 urinary-related events were reported. Bowel-related late AEs of grade 3 or 4 were reported for 26 (3%) patients, 15 (3%) planned for weekly and 11 (2%) daily treatment (Table 4, Table F in S1 Text). For 610 patients with data available at 2 years, grade 3 urinary AEs were reported for 3 (0.5%) and grade 3 bowel AEs for 6 (1%) (Table G in S1 Text). At 4 years, 2/467 (0.4%) patients had grade 3 or 4 bowel toxicity (Table H in S1 Text).

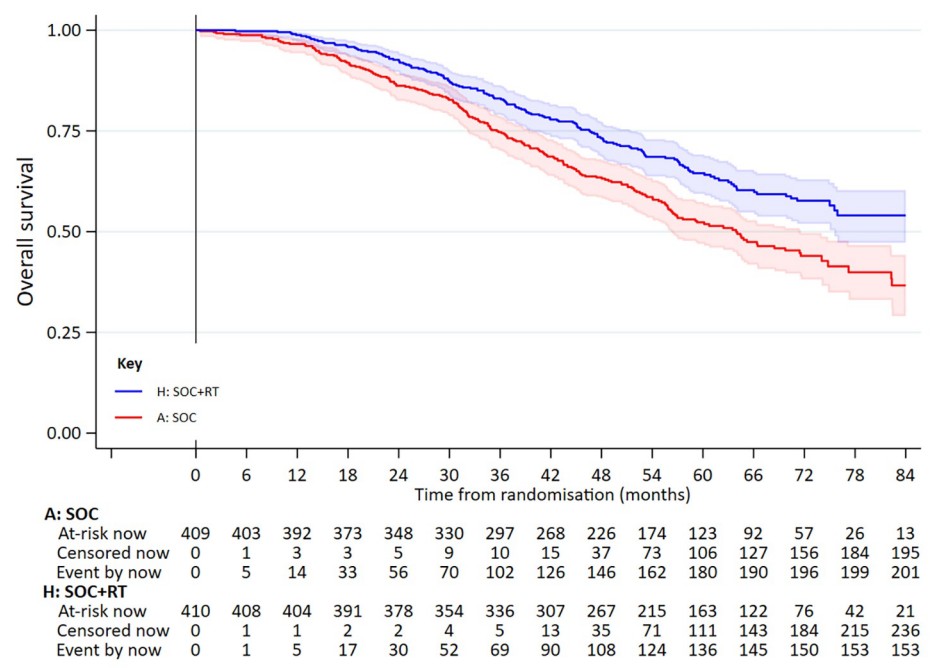

**Fig 3. OS in patients in the low-burden metastatic disease group.** Adjusted HR = 0.64 (95% CI 0.52 to 0.79; $p < 0.001$ [$p = 0.00004$]). HR, hazard ratio; OS, overall survival; RT, radiotherapy to the prostate; SOC, standard of care.

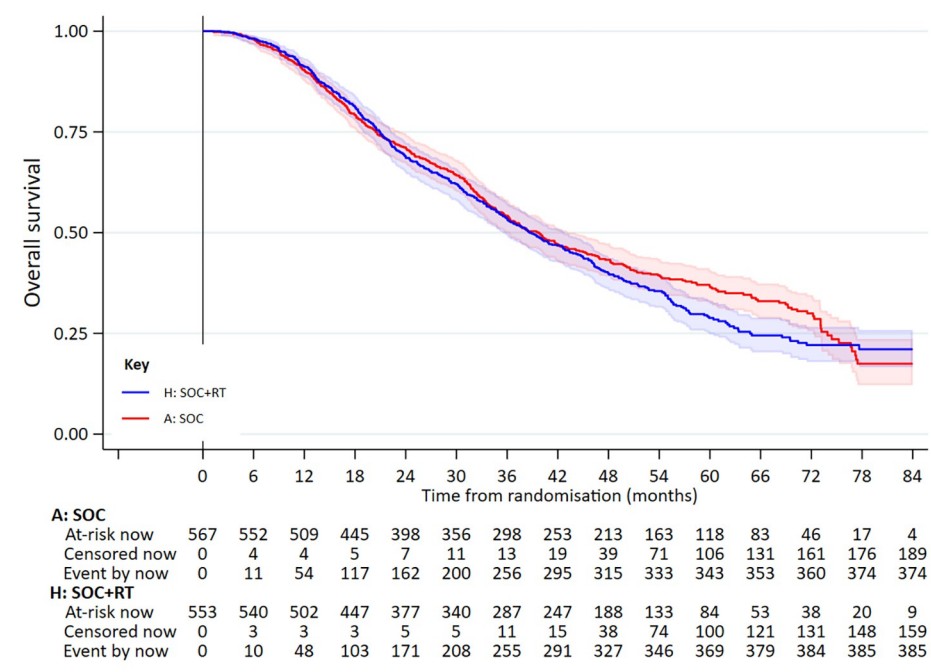

**Fig 4. OS in patients in the high-burden metastatic disease group.** Adjusted HR = 1.11 (95% CI 0.96 to 1.28; $p = 0.164$). HR, hazard ratio; OS, overall survival; RT, radiotherapy to the prostate; SOC, standard of care.

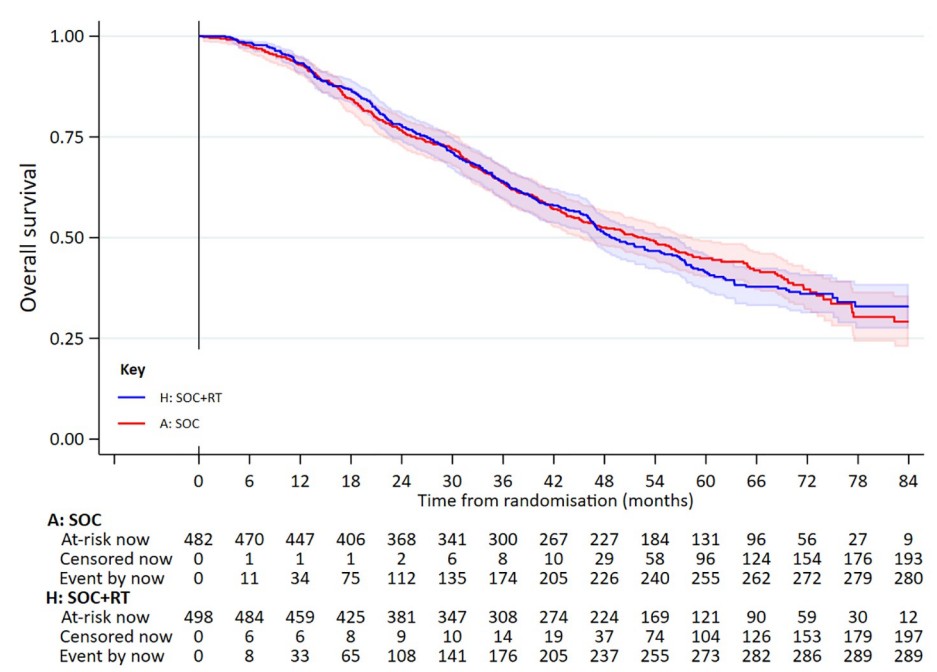

**Fig 5. OS in patients nominated for weekly RT (36 Gy/6 f) prior to randomisation.** Adjusted HR = 1.00 (95% CI 0.85 to 1.18; *p* = 0.974). HR, hazard ratio; OS, overall survival; RT, radiotherapy to the prostate; SOC, standard of care.

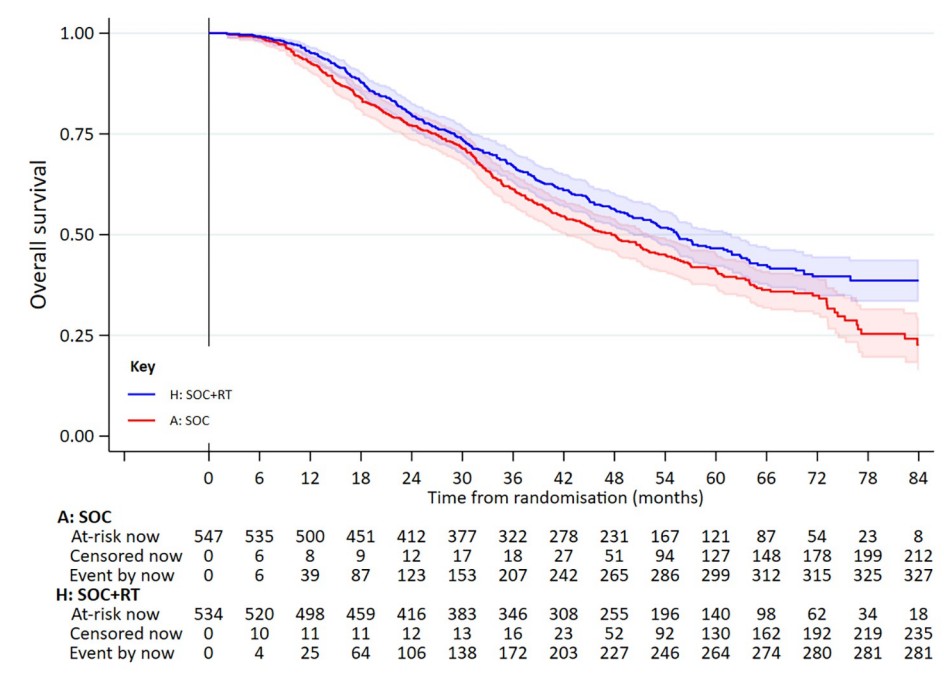

**Fig 6. OS in patients nominated for daily RT (55 Gy/20 f) prior to randomisation.** Adjusted HR = 0.83 (95% CI 0.71 to 0.97; *p* = 0.022). HR, hazard ratio; OS, overall survival; RT, radiotherapy to the prostate; SOC, standard of care.

**Table 3. First symptomatic local event reported (patients with event reported).**

| Type of event | SOC (n = 608) | SOC+RT (n = 601) |
|---|---|---|
| Urinary tract infection | 57 (9%) | 80 (13%) |
| Urinary catheter | 52 (9%) | 44 (7%) |
| Acute kidney injury | 33 (5%) | 34 (6%) |
| TURP | 24 (4%) | 24 (4%) |
| Urinary tract obstruction | 15 (2%) | 15 (3%) |
| Ureteric stent | 19 (3%) | 8 (1%) |
| Nephrostomy | 5 (1%) | 2 (<1%) |
| Colostomy | 3 (<1%) | 3 (1%) |
| Surgery for bowel obstruction | 0 (0%) | 2 (<1%) |
| PCa death | 400 (66%) | 389 (65%) |

PCa, prostate cancer; RT, radiotherapy to the prostate; SOC, standard of care; TURP, transurethral resection of the prostate.

Over the entire reported follow-up period, at least 1 grade 3 to 5 AE was reported for 458 (44%) of SOC and 451 (45%) SOC+RT patients. Areas of focus for this long-term analysis were endocrine disorders: 160/1,052 (15%) SOC versus 155/992 (16%) SOC+RT; musculoskeletal disorders: 112/1,052 (11%) SOC, 104/992 (10%) SOC+RT; blood and bone marrow disorders: 56/1,052 (5%) SOC, 49/992 (5%) SOC+RT; cardiovascular disorders: 46/1,052 (4%) SOC, 56/992 (6%) SOC+RT; renal disorders: 50/1,052 (5%) SOC, 52/992 (5%) SOC+RT; general disorders: 57/1,052 (5%) SOC, 43/992 (4%) SOC+RT; gastrointestinal disorders: 47/1,052 (4%) SOC, 52/992 (5%) SOC+RT; lab abnormalities: 49/1,052 (5%) SOC, 48/992 (5%) SOC+RT (**Table I in S1 Text**, **S7 Fig**). At 2 years, of 715 patients with data available, a grade 3 to 5 AE was reported for 52/320 (16%) SOC and 54/395 (14%) SOC+RT (**Table J in S1 Text**, **S8 Fig**). At 4 years, based on 358 patients, this was 12/133 (9%) SOC versus 29/225 (13%) SOC+RT (**Table K in S1 Text**, **S9 Fig**).

**Table 4. Patients with grade 3/4 worst late RT toxicity score reported over entire time on trial.**

| Toxicity area | SOC+RT | |
|---|---|---|
| | Weekly, 36 Gy/6 f (n = 473) | Daily, 55 Gy/20 f (n = 517) |
| **Urinary** | **10 (2%)** | **10 (2%)** |
| Hematuria | 4 (1%) | 4 (1%) |
| Urethral stricture | 3 (1%) | 4 (1%) |
| Cystitis | 3 (1%) | 4 (1%) |
| **Bowel** | **15 (3%)** | **11 (2%)** |
| Proctitis | 9 (2%) | 5 (1%) |
| Diarrhea | 6 (1%) | 6 (1%) |
| Rectal–anal stricture | 0 (0%) | 0 (0%) |
| Rectal ulcer | 0 (0%) | 1 (<1%) |
| Bowel obstruction | 1 (<1%) | 1 (<1%) |

Note: SOC+RT in safety population (RTOG scale; patients with RT started within 1 year of randomisation). There were no reported grade 5 late RT toxicity events.

RT, radiotherapy to the prostate; SOC, standard of care.

**Table 5. Summary of QoL analyses.**

| Outcome measure | Analysis | Average over first 2 years on trial | | Difference (95% CI) |
|---|---|---|---|---|
| | | SOC | SOC+RT | |
| **Global QoL (%)** | Partly conditional | 73.2% | 72.4% | −0.8% (−2.5% to 0.9%) |
| | Composite outcome | 60.3% | 61.6% | 1.3% (−1.1% to 3.8%) |
| | Cross-sectional: 12 weeks | n/a | n/a | −2.9% (−4.8% to −1.0%) |
| | Cross-sectional: 24 weeks | n/a | n/a | −0.9% (−3.1% to 1.3%) |
| | Cross-sectional: 60 weeks | n/a | n/a | −1.4% (−4.1% to 1.3%) |
| | Cross-sectional: 104 weeks | n/a | n/a | 1.8% (−2.4% to 6.0%) |
| **QLQ-30 Summary Score (%)** | Partly conditional | 85.4% | 84.2% | −1.2% (−2.4% to 0.0%) |
| | Composite outcome | 70.6% | 71.7% | 1.2% (−1.3% to 3.6%) |
| | Cross-sectional: 12 weeks | n/a | n/a | −2.0% (−3.2% to −0.8%) |
| | Cross-sectional: 24 weeks | n/a | n/a | −1.0% (−2.3% to 0.4%) |
| | Cross-sectional: 60 weeks | n/a | n/a | −1.0% (−2.8% to 0.7%) |
| | Cross-sectional: 104 weeks | n/a | n/a | 0.9% (−1.8% to 3.6%) |

Note: Partly conditional estimates are based on observed and multiply imputed data from patients alive at scheduled assessments within the first 2 years since randomisation. Composite outcome estimates are based on observed data and implied imputation of missing data from scheduled assessments when a patient was alive, and the assumption of a patient's Global QoL/QLQ-30 Summary Score being 0% at all scheduled assessments after they have died. Cross-sectional analyses estimate the difference in average Global Qol/QLQ-30 Summary Score between SOC+RT and SOC treatment groups at the specified scheduled assessment, controlling for response at baseline, in complete cases only (i.e., in patients with outcome data provided at baseline and who have survived and for who outcome data is available at the specified scheduled assessment).

QoL, quality of life; RT, radiotherapy to the prostate; SOC, standard of care.

## QoL

There was no evidence of a difference in QoL scores over time between the allocated treatment groups. Average Global QoL in the first 2 years after randomisation across all patients was 73.2% SOC and 72.4% SOC+RT; absolute difference −0.8% (95% CI −2.5% to 0.9%), $p = 0.349$ (partly conditional analysis) (**Table 5**, **S3 Fig**). When including patients who had died prior to an assessment as having a Global QoL score of 0% at that assessment, average Global QoL was 60.3% SOC versus 61.6% SOC+RT; absolute difference 1.3% (95% CI -1.1% to 3.8%), $p = 0.287$ (composite outcome analysis) (**Table 5**, **S4 Fig**).

Average QLQ-30 Summary Score in the first 2 years across all patients was 85.4% SOC and 84.2% SOC+RT; absolute difference −1.2% (95% CI −2.4% to 0.0%), $p = 0.050$ (partly conditional analysis) (**Table 5**, **S5 Fig**). When assuming a value of 0% for assessments after a patient had died, average Summary Score was 70.6% SOC and 71.7% SOC+RT; absolute difference 1.2% (95% CI −1.3% to 3.6%), $p = 0.365$ (composite outcome analysis) (**Table 5**, **S6 Fig**).

Cross-sectional analyses of both Global QoL and QLQ-30 Summary Score indicated evidence of poorer QoL at week 12 after randomisation for patients allocated to SOC+RT— Global QoL absolute difference −2.9% (95% CI −4.8% to −1.0%, $p = 0.003$); Summary Score absolute difference −2.0% (95% CI −3.2% to −0.8%, $p = 0.001$)—but not at other assessments (**Table 5**).

## Discussion

This final analysis has confirmed that prostate RT improves OS in men with newly diagnosed, low-burden metastatic prostate cancer, but not in men with high-burden disease. The magnitude of the survival benefit is substantial and clinically relevant, particularly given that prostate RT is a relatively cheap, widely accessible, and well-tolerated treatment.

These results of the final analysis confirm the findings from the initial analysis. The additional 2 years of follow-up, and the subsequent increase in the number of events for analysis, has reduced the CIs around the point estimate of the HR of the OS benefit for prostate RT. However, the point estimate itself has changed very little, improving from 0.68 to 0.64 for men in the low metastatic disease risk group. This result is consistent with that from the smaller HORRAD trial [10]. Our new data strongly support those guidelines already recommending the use of prostate RT in men with low-burden metastatic disease. We have not found any benefit for prostate RT in men with high-burden disease, either in OS or in preventing interventions for local disease progression.

We found no compelling evidence of a difference in efficacy or toxicity between the 2 RT dose-schedules tested. The weekly schedule of 36 Gy in 6 fractions over 6 weeks has an obvious practical advantage in terms of convenience and may be preferred for that reason. A daily schedule might be preferred if pelvic nodal RT were to be used in addition or if RT dose escalation was thought to be appropriate. Prostate RT did not have any long-term impact on QoL either in this trial, or in the HORRAD trial [11]. The risk of toxicity from prostate RT, although low, could be further reduced by the use of more contemporary intensity modulated techniques [12].

The criteria used in the trial to classify cases as low or high burden were taken from those used in the CHAARTED trial [5]. These criteria are based on the presence or absence of visceral disease on CT scan, together with the number and the location of bone metastases on bone scan. Patients with only lymph node metastases have low-burden disease, regardless of the extent of nodal disease. There is no good reason to think that these criteria are optimal for identifying those patients with metastatic disease who stand to benefit from prostate RT. The initial analysis of STAMPEDE suggested that the survival benefit from prostate RT gradually decreased in magnitude as the number of bone metastases visible on a baseline bone scan increased [13]. One could decide to identify patients suitable for prostate RT based solely on the number of bone metastases visible on baseline bone scan, regardless of location. A count-of-metastases approach would be simpler to use in the clinic than the CHAARTED definition and would likely increase the number of men considered suitable for prostate RT.

The trial has several strengths, including the randomised design, the large number of events for analysis, and recruitment from over 100 centres, which adds to the generalisability of the results. The main limitations of the study are the changes in clinical practice since the trial started, particularly with regard to imaging techniques and systemic treatment. The trial recruited between 2013 and 2016 and, while this has the benefit of long follow-up, it also means that newer imaging techniques, such as PSMA (Prostate-specific membrane antigen) PET and whole body MRI, were unavailable. It is important to note that low-burden disease in the trial was defined according to bone scan and CT scan. There is no agreed definition of metastatic disease burden based solely on PSMA PET or on whole body MRI. In patients without visceral disease but who have more than 4 bone metastases on PET or MRI, a bone scan may be required in addition, in order to determine suitability for prostate RT. If this is not practicable, and there remains uncertainty as to whether a patient has high- or low-burden disease, there is a strong argument for using prostate RT.

The systemic treatment of metastatic prostate cancer has changed since the trial recruited. Standard treatment for men with low-burden metastatic disease now includes one of the newer hormone agents (abiraterone or apalutamide or enzalutamide) in addition to ADT. The effect of these agents on the survival benefit of prostate RT is unknown. Similarly, the effect of prostate RT on the survival benefit of the newer hormonal agents is also unknown. Based on current evidence, it is reasonable to assume that both prostate RT and one of the newer hormonal agents should be considered SOC for low-burden metastatic disease, in addition to

ADT. The PEACE-1 Trial is testing the use of prostate RT in men receiving ADT + abiraterone.

In summary, this final analysis confirms that prostate RT improves OS in men with low-burden, newly diagnosed, metastatic prostate cancer, indicating that it should be recommended as a SOC.

## Supporting information

**S1 Fig. SLEFS in all patients.** Adjusted HR = 1.00 (95% CI 0.90 to 1.13; $p$ = 0.931). HR, hazard ratio; RT, radiotherapy to the prostate; SLE, symptomatic local event; SOC, standard of care.
(TIF)

**S2 Fig. LIFS in all patients.** Adjusted HR = 0.94 (95% CI 0.83 to 1.06; $p$ = 0.286). HR, hazard ratio; LI, local intervention; RT, radiotherapy to the prostate; SOC, standard of care.
(TIF)

**S3 Fig. Model-estimated Global QoL (partly conditional analysis in all patients).** Difference in weighted average: −0.8% (95% CI −2.5% to 0.9%; $p$ = 0.349). QoL, quality of life; RT, radiotherapy to the prostate; SOC, standard of care.
(TIF)

**S4 Fig. Model-estimated Global QoL (composite outcome analysis in all patients).** Difference in weighted average: 1.3% (95% CI −1.1% to 3.8%; $p$ = 0.287). QoL, quality of life; RT, radiotherapy to the prostate; SOC, standard of care.
(TIF)

**S5 Fig. Model-estimated QLQ-30 Summary Score (partly conditional analysis in all patients).** Difference in weighted average: −1.2% (95% CI −2.4% to 0.0%; $p$ = 0.050). RT, radiotherapy to the prostate; SOC, standard of care.
(TIF)

**S6 Fig. Model-estimated QLQ-30 Summary Score (composite outcome analysis in all patients).** Difference in weighted average: 1.2% (95% CI −1.3% to 3.6%; $p$ = 0.365). RT, radiotherapy to the prostate; SOC, standard of care.
(TIF)

**S7 Fig. Highest grade AE reported over entire time on trial (CTCAE v4.0, all patients).** AE, adverse event; CTCAE, Common Terminology Criteria for Adverse Events; RT, radiotherapy to the prostate; SOC, standard of care.
(TIF)

**S8 Fig. Highest grade AE reported at 2 years in patients prior to disease progression (CTCAE v4.0).** AE, adverse event; CTCAE, Common Terminology Criteria for Adverse Events; RT, radiotherapy to the prostate; SOC, standard of care.
(TIF)

**S9 Fig. Highest Grade AE reported at 4 years in patients prior to disease progression (CTCAE v4.0).** AE, adverse event; CTCAE, Common Terminology Criteria for Adverse Events; RT, radiotherapy to the prostate; SOC, standard of care.
(TIF)

**S10 Fig. Time to reported initiation of abiraterone or enzalutamide from randomisation.** RT, radiotherapy to the prostate; SOC, standard of care.
(TIF)

**S11 Fig. Time to reported initiation of abiraterone or enzalutamide from FFS event.** FFS, failure-free survival; RT, radiotherapy to the prostate; SOC, standard of care.
(TIF)

**S1 Text. Table A in S1 Text. Baseline characteristics for metastatic volume analyses.** ADT, androgen deprivation therapy; IQR, interquartile range; PSA, prostate specific antigen; RT, radiotherapy to the prostate; SOC, standard of care; WHO, World Health Organization. **Table B in S1 Text. Eligibility status following participant audit.** RT, radiotherapy to the prostate; SOC, standard of care. **Table C in S1 Text. Sensitivity analyses on OS based on explicit eligibility.** ITT, intention-to-treat; OS, overall survival; RT, radiotherapy to the prostate; SOC, standard of care. **Table D in S1 Text. First local intervention event reported (patients with event reported).** PCa, prostate cancer; RT, radiotherapy to the prostate; SOC, standard of care; TURP, transurethral resection of the prostate. **Table E in S1 Text. Summary of analyses of time to local event outcomes**. *Subdistribution HR for competing risks models. ^Cox model, adjusting for age, nodal involvement, WHO performance status, regular aspirin or NSAID use and planned SOC docetaxel at randomisation, stratified by randomisation time period. +Fine and Gray model with outcome excluding PCa death and death from any cause as competing risk. NSAID, nonsteroidal anti-inflammatory drug; PCa, prostate cancer; SOC, standard of care; WHO, World Health Organization. **Table F in S1 Text. Grade 3 to 5 late RT toxicities reported over entire time on trial (RTOG).** Note: Treatment arms correspond to safety population; patients with ≥1 Follow-Up CRF returned. RT, radiotherapy to the prostate; RTOG, Radiation Therapy Oncology Group; SOC, standard of care. **Table G in S1 Text. Grade 3 to 5 late RT toxicities reported at 2 years (RTOG).** Note: Treatment arms correspond to safety population; patients with ≥1 Follow-Up CRF returned and no reported progression at 2 years. RT, radiotherapy to the prostate; RTOG, Radiation Therapy Oncology Group; SOC, standard of care. **Table H in S1 Text. Grade 3 to 5 late RT toxicities reported at 4 years (RTOG).** Note: Treatment arms correspond to safety population; patients with ≥1 Follow-Up CRF returned and no reported progression at 4 years. RT, radiotherapy to the prostate; RTOG, Radiation Therapy Oncology Group; SOC, standard of care. **Table I in S1 Text. Grade 3 to 5 AEs reported over entire time on trial, overall and for selected body systems (CTCAE).** Note: Treatment arms correspond to safety population; patients with ≥1 Follow-Up/SAE CRF returned. AE, adverse event; RT, radiotherapy to the prostate; RTOG, Radiation Therapy Oncology Group; SOC, standard of care. **Table J in S1 Text. Grade 3 to 5 AEs reported at 2 years, overall and for selected body systems (CTCAE).** Note: Treatment arms correspond to safety population; patients with ≥1 Follow-Up/SAE CRF returned and no reported progression at 2 years. AE, adverse event; CTCAE, Common Terminology Criteria for Adverse Events; RT, radiotherapy to the prostate; SOC, standard of care. **Table K in S1 Text. Grade 3 to 5 AEs reported at 4 years, overall and for selected body systems (CTCAE).** Note: Treatment arms correspond to safety population; patients with ≥1 Follow-Up/SAE CRF returned and no reported progression at 4 years. AE, adverse event; CTCAE, Common Terminology Criteria for Adverse Events; RT, radiotherapy to the prostate; SOC, standard of care.
(DOCX)

**S2 Text. Statistical analysis plan.**
(PDF)

**S3 Text. CONSORT checklist.** CONSORT, Consolidated Standards of Reporting Trials.
(PDF)

**S4 Text. List of investigators, oversight committees, and contributors.**
(PDF)

## Acknowledgments

Large-scale trials do not happen without huge collaborations. Thanks to all central and site staff who have made the STAMPEDE trial happen. See **S4 Text** for full list of investigators, oversight committees, and contributors. In particular, thanks to all the people who have chosen participate in STAMPEDE and their families and friends who have supported them.

The views expressed are those of the authors and not necessarily those of the NIHR or the Department of Health and Social Care.

### Investigators and collaborators

See Credit List included as **S4 Text** and on the STAMPEDE trial website: http://www.stampedetrial.org/media-section/presentation-repository/trial-recognition.

## Author Contributions

**Conceptualization:** Chris C. Parker, Nicholas D. James, Noel W. Clarke, Peter Robson, Mahesh K. B. Parmar, Matthew R. Sydes.

**Data curation:** Chris C. Parker, Nicholas D. James, Christopher D. Brawley, Noel W. Clarke, Adnan Ali, Claire L. Amos, Gerhardt Attard, Simon Chowdhury, Adrian Cook, William Cross, David P. Dearnaley, Duncan C. Gilbert, Clare Gilson, Silke Gillessen, Alex Hoyle, Rob J. Jones, Ruth E. Langley, Zafar I. Malik, Malcolm D. Mason, David Matheson, Robin Millman, Mary Rauchenberger, Hannah Rush, J Martin Russell, Hannah Sweeney, Amit Bahl, Alison Birtle, Lisa Capaldi, Omar Din, Daniel Ford, Joanna Gale, Ann Henry, Peter Hoskin, Mohammed Kagzi, Anna Lydon, Joe M. O'Sullivan, Sangeeta A. Paisey, Omi Parikh, Delia Pudney, Vijay Ramani, Peter Robson, Narayanan Nair Srihari, Jacob Tanguay, Mahesh K. B. Parmar, Matthew R. Sydes.

**Formal analysis:** Christopher D. Brawley, Adrian Cook, Matthew R. Sydes.

**Funding acquisition:** Chris C. Parker, Noel W. Clarke, David P. Dearnaley, Malcolm D. Mason, Mahesh K. B. Parmar, Matthew R. Sydes.

**Investigation:** Chris C. Parker, Nicholas D. James, Christopher D. Brawley, Noel W. Clarke, Adnan Ali, Claire L. Amos, Gerhardt Attard, Adrian Cook, William Cross, David P. Dearnaley, Hassan Douis, Duncan C. Gilbert, Clare Gilson, Silke Gillessen, Alex Hoyle, Rob J. Jones, Ruth E. Langley, Zafar I. Malik, Malcolm D. Mason, David Matheson, Robin Millman, Mary Rauchenberger, Hannah Rush, J Martin Russell, Hannah Sweeney, Amit Bahl, Alison Birtle, Lisa Capaldi, Omar Din, Daniel Ford, Joanna Gale, Ann Henry, Peter Hoskin, Mohammed Kagzi, Anna Lydon, Joe M. O'Sullivan, Sangeeta A. Paisey, Omi Parikh, Delia Pudney, Vijay Ramani, Peter Robson, Narayanan Nair Srihari, Jacob Tanguay, Mahesh K. B. Parmar, Matthew R. Sydes.

**Methodology:** Chris C. Parker, Nicholas D. James, Noel W. Clarke, Malcolm D. Mason, Mahesh K. B. Parmar, Matthew R. Sydes.

**Project administration:** Chris C. Parker, Nicholas D. James, Christopher D. Brawley, Noel W. Clarke, Adnan Ali, Claire L. Amos, Clare Gilson, Alex Hoyle, Hannah Rush, Hannah Sweeney, Mahesh K. B. Parmar, Matthew R. Sydes.

**Software:** Mary Rauchenberger.

**Supervision:** Chris C. Parker, Nicholas D. James, Noel W. Clarke, Claire L. Amos, Gerhardt Attard, Simon Chowdhury, Adrian Cook, William Cross, David P. Dearnaley, Duncan C.

Gilbert, Silke Gillessen, Rob J. Jones, Ruth E. Langley, Zafar I. Malik, David Matheson, Robin Millman, J Martin Russell, Mahesh K. B. Parmar, Matthew R. Sydes.

**Validation:** Christopher D. Brawley, Noel W. Clarke, Adnan Ali, Adrian Cook, Alex Hoyle, Matthew R. Sydes.

**Visualization:** Chris C. Parker, Christopher D. Brawley, Adrian Cook, Mahesh K. B. Parmar, Matthew R. Sydes.

**Writing – original draft:** Chris C. Parker, Nicholas D. James, Christopher D. Brawley, Noel W. Clarke, Adrian Cook, Matthew R. Sydes.

**Writing – review & editing:** Chris C. Parker, Nicholas D. James, Christopher D. Brawley, Noel W. Clarke, Adnan Ali, Claire L. Amos, Gerhardt Attard, Simon Chowdhury, Adrian Cook, William Cross, David P. Dearnaley, Duncan C. Gilbert, Clare Gilson, Silke Gillessen, Alex Hoyle, Ruth E. Langley, Zafar I. Malik, Malcolm D. Mason, David Matheson, Robin Millman, Mary Rauchenberger, Hannah Rush, J Martin Russell, Hannah Sweeney, Amit Bahl, Alison Birtle, Lisa Capaldi, Omar Din, Daniel Ford, Joanna Gale, Ann Henry, Peter Hoskin, Mohammed Kagzi, Anna Lydon, Joe M. O'Sullivan, Sangeeta A. Paisey, Omi Parikh, Delia Pudney, Vijay Ramani, Peter Robson, Narayanan Nair Srihari, Jacob Tanguay, Mahesh K. B. Parmar, Matthew R. Sydes.

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
