## [Editor Report · Decision Letter 0]

6 Jan 2022

Dear Dr Sydes, 

Thank you for submitting your manuscript entitled "Radiotherapy to the prostate for men with metastatic prostate cancer: long-term results from the STAMPEDE randomised controlled trial (NCT00268476)" for consideration by PLOS Medicine.

Your manuscript has now been evaluated by the PLOS Medicine editorial staff and I am writing to let you know that we would like to send your submission out for external peer review.

Please re-submit your manuscript within two working days, i.e. by Jan 10 2022 11:59PM.

Kind regards,

Callam Davidson

Associate Editor

PLOS Medicine

---

## [Decision Letter · Decision Letter 1]

7 Feb 2022

Dear Dr. Sydes,

Thank you very much for submitting your manuscript "Radiotherapy to the prostate for men with metastatic prostate cancer: long-term results from the STAMPEDE randomised controlled trial (NCT00268476)" (PMEDICINE-D-22-00022R1) for consideration at PLOS Medicine. 

[LINK]

In light of these reviews, I am afraid that we will not be able to accept the manuscript for publication in the journal in its current form, but we would like to consider a revised version that addresses the reviewers' and editors' comments. Obviously we cannot make any decision about publication until we have seen the revised manuscript and your response, and we plan to seek re-review by one or more of the reviewers. 

We hope to receive your revised manuscript by Feb 28 2022 11:59PM. Please email us (plosmedicine@plos.org) if you have any questions or concerns.

We look forward to receiving your revised manuscript. 

Sincerely,

Callam Davidson, 

PLOS Medicine

plosmedicine.org

In the Data Availability Statement (in the submission form) please include an appropriate contact email address for inquiries (this cannot be a study author).

Please remove the trial registration number from your title. 

In your abstract ‘Methods and Findings’ please make the following changes:

* Please indicate the dates during which study enrollment and follow up occurred.

* Please include the duration of follow-up.

* Please clearly define the study outcomes. 

* Please state that the analysis was intention-to-treat. 

* Please include the important variables adjusted for in the analysis.

* Please state that there was no blinding to treatment allocation. 

* Please include the important stratification factors adjusted for in the analysis.

* Please include a brief summary of adverse events. 

* Please include a single sentence summary of the study’s main limitations at the end of the section (beginning ‘The main limitations of the study were…’, or similar).

Please remove details of funding from the end of the abstract.

Please update citations so that they are not superscript, are within square brackets, and precede punctuation. 

Please provide the name of the relevant Institutional Review Board/s that provided ethics approval in the Methods. 

Thank you for providing a CONSORT checklist. Please do the following:

* Update the checklist to use section names and paragraph numbers rather than page numbers (which change during the revision process). 

* Cite the checklist in your methods (e.g. ‘This trial is reported per the Consolidated Standards of Reporting Trials (CONSORT; see S1 checklist)’ or similar). 

Please report p values to 3 decimal places (p<0.001) (e.g. at lines 207 and 218, but please check throughout).

Please define abbreviations used in your Tables and Figures (in the legend or in the figure/table itself where appropriate).

Please provide the unadjusted comparisons as well as the adjusted comparisons in Table 2. Please also indicate factors adjusted for in the legend. 

Line numbering appears to disappear after page 17, please correct this to be continuous throughout the document. 

In your Discussion, please include a paragraph covering the strengths and limitations of the study. 

Author contributions and Competing interests can be removed from the manuscript as this information is captured as part of the submission form. 

Please remove any formatting from your references (e.g. bold/italics) and only use et al. after listing the first six authors. 

Comments from the reviewers:

Reviewer #1: Parker et al present the updated results of the STAMPEDE trial, arm H. This trial randomized >2000 men with newly diagnosed metastatic prostate cancer in the UK and Switzerland to standard of care (lifelong androgen deprivation therapy) +/- radiation therapy (RT) to the prostate. After the start of enrollment, it was determined that results would be analyzed by the volume of metastatic disease. The authors had previously reported in 2018 that prostate RT improves survival in men with low-volume disease but not high-volume disease, with a significant interaction (Parker et al, Lancet 2018). In this updated analysis, they demonstrate that the survival benefit differences found previously persist with additional follow-up. Furthermore, they present data on local events in the forms of Local Intervention Free Survival and Symptomatic Local Event-Free Survival. 

This is an important update of the STAMPEDE trial, arm H, which helped change the treatment paradigm for men with low-volume de novo metastatic prostate cancer. While the updated survival findings, which demonstrated the continued benefit of prostate RT in low-volume disease, are not surprising, the authors do provide additional results in terms of local event analyses, toxicity including from RT, and quality of life (QOL) metrics. The findings of this study are important for the general medical public given the prevalence of prostate cancer. The manuscript is generally well-written and presents appropriate analyses.

Major comments

- Local Intervention Free Survival (LIFS) was defined as "consisting of time from randomisation to the first report on case report forms of TURP, ureteric stent, surgery for bowel obstruction, urinary catheter, nephrostomy, colostomy, death from prostate cancer" (lines 120-122). I am not confident that death from prostate cancer should be considered in the definition of LIFS, since death from prostate cancer is uncommonly a consequence of local progression but rather typically due to an increase in metastatic disease burden. Did the authors consider an analysis of LIFS using a definition which excludes death from prostate cancer, using death from any cause as a competing risk event? Similarly, did the authors consider an analysis of Symptomatic Local Event-Free Survival (SLEFS) using a definition that excludes death from prostate cancer, again using competing risks survival analysis? Understandably, the sample sizes for these analyses are likely to be small since it appears that the majority of patients did not have events besides death from prostate cancer be reported. 

- The authors found that when stratified by RT schedule, the hazard ratio (HR) for weekly RT was 1.00 and for daily RT was 0.83 (interaction p=0.088). They then looked at specifically at patients with low-volume status and found a HR of 0.67 for weekly RT and 0.62 for daily RT (interaction p=0.732), demonstrating that RT schedule does not appear to influence outcomes for low-volume patients. As additional exploratory analysis, it may be informative to see a similar analysis of HRs by fractionation schedule for high-volume patients. 

- One possible confounder in this study is the receipt of additional therapies not reported here, namely novel androgen receptor signaling inhibitors (ARSI). It is possible that patients in one group were more likely to have received ARSIs which can influence multiple outcomes, including survival, local events, toxicities, and quality of life metrics. Are the authors able to provide data on use of these novel therapies and if not, they should discuss this as a limitation of the study.

Minor comments

- Line 155: There is an extra period after "criteria."

- In Figure 1, it is unclear to me how the n's in the Long-term analysis row were derived from the Follow-up row, e.g. for the first column, 333+74+609=1016, yet the diagram states that n=1029 were analyzed for efficacy. Can the authors clarify this?

- It might be helpful to relabel metastatic disease burden as "low" and "high" in Table 1 to be uniform in the terminology.

- The legend descriptions for Figures 2A and 2B appear flipped (lines 210-218). There is no legend for Figure 2C.

- The authors state that they performed "cause-specific and competing-risk analyses." They should clarify whether this was done using Cox proportional hazard regression vs Fine-Gray regression and define what the competing risks were. Additionally, for prostate cancer-specific mortality, SLEFS, and LIFS, it would be helpful to see interaction p-values as well.

- The authors should clarify the percentage of patients who received 3D conformal RT (3D RT) vs intensity-modulated RT (IMRT). My sense is that the majority of patients likely received 3D RT which, in combination with the generous margins used (8mm posterior and 10mm elsewhere), is expected to increase the level of toxicity from RT compared to more modern, albeit expensive, techniques (i.e. image-guided IMRT with tighter margins, usually around 5mm if not less) used in many other parts of the world. This complicates the interpretation of toxicity and quality of life data presented here and hence should be discussed as a limitation.

Reviewer #2: This paper reports on the long term overall survival, quality of life and local events in patients with primary metastatic prostate cancer, randomised between with radiotherapy to the prostate or standard of care in the SRAMPEDE trial. This is an essential trial since it altered the guidelines into advising radiotherapy to the prostate in patients with low volume metastatic disease. The presented update on the survival data that confirms previous findings is very important. Also reporting on local problems and quality of life is useful, as local radiotherapy of the prostate is known to have a substantial impact on patients' health-related-quality-of-life potentially. The manuscript is well written and has a clear message. 

There are a few issues in this report that should be addressed.

1) In the introduction, the authors write that they report on data on freedom from local interventions and mention urinary catheter, ureteric stents, nephrostomies, colostomy. Why did the authors eventually include death in their analysis on symptomatic local event? This is already addressed in their survival analysis. It there a difference in time to local event or proportion of local events when death is not included in this analysis? 

2) Table 3 misses in the manuscript.

3) In the introduction, the authors write that the first report on quality of life of this trial. The HORRAD group also published results on the effect on HrQoL of radiotherapy in patients with metastastic prostate cancer. It would be helpful to compare the results of both similar trials in the discussion.

4) In analysing QoL questionnaires, only the global QoL scores were presented. Were there any differences in the different domains of the questionnaires between the two groups? 

5) For time-to-event variables, report the number of events but not the proportion. Avoid reporting the percentage of deaths in reporting on survival. Although it is important to report the number of events, patients entered the study at different times and were followed for different periods; hence, the reported proportion is meaningless.

Reviewer #3: This is a useful and well-conducted RCT on the long-term results of radiotherapy on the prostate for men with metastatic prostate cancer. The study design, statistical methods and analyses, and presentation (tables and figures) and interpretation of the results are mostly adequate and of a good standard. However, there are still a few issues needing attention.

1) For the subgroup analyses on low and high burden groups, there are 6% missing data (without disease burden classification). But I haven't seen any sensitivity analysis on the missing data. What is the impact of missing data on the analysis results? Any potential bias? At least we'd like to see the patterns and characteristics of the patients without burden classification as compared to those with burden classification. May need to address the issue as a limitation in the discussion if missing is not at random.

2) In the analysis of the high-burden metastatic disease group (figue 2B), significant interaction was found. However, it is not clear whether the interaction term was dealt with in the analysis, and if yes, how it was dealt with. This question also applies to a few other analyses involving significant interations terms.

3) Sample size calculation was never mentioned in the paper. I am aware this is a follow-up on the previous published paper but at least authors can mention the sample size and direct readers to previous papers or protocol.

4) In statistical analysis section on page 11, it says "The proportional hazards assumption was tested", but with what method?

[LINK]

---

## [Decision Letter · Decision Letter 2]

1 Apr 2022

Dear Dr. Sydes,

Thank you very much for re-submitting your manuscript "Radiotherapy to the prostate for men with metastatic prostate cancer: long-term results from the STAMPEDE randomised controlled trial (NCT00268476)" (PMEDICINE-D-22-00022R2) for review by PLOS Medicine.

I have discussed the paper with my colleagues and it was also seen again by three reviewers. I am pleased to say that provided the remaining editorial and production issues are dealt with we are planning to accept the paper for publication in the journal.

[LINK]

We look forward to receiving the revised manuscript by Apr 08 2022 11:59PM.   

Sincerely,

Callam Davidson, 

Associate Editor 

PLOS Medicine

plosmedicine.org

Requests from Editors:

Please update your title to include the setting (‘Radiotherapy to the prostate for men with metastatic prostate cancer in the UK and Switzerland: long-term results from the STAMPEDE randomised controlled trial’).

As your author list runs to >30 authors, please consider grouping authors under a collaborative group such that the named author list is ≤30 authors. 

Data Availability Statement: Thank you for providing a contact email address (mrcctu.datareleaserequest@ucl.ac.uk). Please ensure this is included in your Data Availability Statement (via the submission form), as the current version recommends contacting the trial team (PLOS does not permit listing study authors as the point of contact for data requests).

In your abstract ‘Methods and Findings’ please make the following changes:

* Please include the setting.

* Please include the length of follow up

* Per our in-house style, please include a single sentence summary of the study’s main limitations at the end of the section (beginning ‘The main limitations of the study were…’, or similar). See previous PLOS Medicine papers for examples.

Please ensure the Abstract and Title are both updated in the submission form as well as the main paper (currently the two versions differ). 

Please report p values to 3 decimal places throughout the document (P<0.001 should be the minimum value reported). Example at line 27 and in Figure 2A, but please check throughout. 

There is no need to include the original p values as footnotes.

Line 35: Please update to ‘indicating that it should be recommended as a standard of care’ (applies also to concluding paragraph).

Citations throughout should precede punctuation throughout.

Your Author Summary should follow the format here: https://journals.plos.org/plosmedicine/s/revising-your-manuscript#loc-author-summary

Please update your CONSORT checklist to use section names and paragraph numbers rather than page numbers (which change during the revision process). 

The name of the CONSORT checklist file also seems like it does not match the manuscript (‘The Impact of a Community-Oriented Problem-Based Learning Curriculum Reform on the Quality of Primary Care Delivered by Graduates’). Please check.

Thank you for presenting your unadjusted comparisons in Table 2. A line can be added to the Methods section noting that these comparisons were presented in response to a request from the handling editor and thus represents a change from the SAP. 

The Financial Disclosure can be removed from the main text, the relevant information should instead be captured in the submission form (answers will then be published as metadata).

Please remove the abbreviations list from the main text and instead define abbreviations on first use/in table and figure legends. 

Line 88: Please update the citations here to [2, 3]. 

When presenting hazard ratios in the Results, please consistently indicate whether these are adjusted or unadjusted ratios (e.g., Section 3.3, paragraph 1 and paragraph 2, HRs are reported differently despite both representing adjusted HRs). 

Apologies if I missed it but I couldn’t find an in-text reference for Figure S5?

Comments from Reviewers:

Reviewer #1: I would like to thank the authors for appropriately addressing the comments which I previously raised. I do not have any additional major comments. The authors should be congratulated on this important contribution. 

Of note, Table S5 appears to be currently formatted in such a way that the right portion of the table is cut off and not visible in either the PDF or the Word document. I would recommend that the orientation of this page be changed to landscape to allow for the entire table to be viewed. 

Reviewer #2: The authors made several improvements to the document based on the reviewers comments, I have no further issues to raise.

Reviewer #3: Thanks authors for their effort to improve the manuscript. I am satisfied with the response and revision. No further issues needing attention.

[LINK]

---

## [Editor Report · Decision Letter 3]

22 Apr 2022

Dear Dr Sydes, 

On behalf of my colleagues and the Academic Editor, Dr Brenton, I am pleased to inform you that we have agreed to publish your manuscript "Radiotherapy to the prostate for men with metastatic prostate cancer in the UK and Switzerland: long-term results from the STAMPEDE randomised controlled trial" (PMEDICINE-D-22-00022R3) in PLOS Medicine.

Prior to final acceptance, please make the following changes:

Early in the "Methods and findings" subsection of the abstract, please add a sentence, say, to quote the demographic characteristics of study participants. 

There appears to have been a small misunderstanding on one editorial point: Please add a new final sentence to the "Methods and findings" subsection of the abstract, which should begin "Study limitations include ..." or similar and should quote 2-3 of the study's main limitations. 

Please adapt the "Author Summary" so that each of the three subsections consists of no more than 3-4 points. To achieve this, some points can be removed (e.g., "Prostate cancer is the most common cancer ..." and "The dataset was frozen in 2021 ...") and in some cases shorter points could be combined. 

We suggest adapting "CONSORT diagram" to "Participant flowchart" throughout. 

PRESS

Sincerely, 

Richard Turner PhD, for Callam Davidson 

rturner@plos.org